# Molecular Mechanisms of Alcohol-Induced Colorectal Carcinogenesis

**DOI:** 10.3390/cancers13174404

**Published:** 2021-08-31

**Authors:** Caroline H. Johnson, Jaya Prakash Golla, Evangelos Dioletis, Surendra Singh, Momoko Ishii, Georgia Charkoftaki, David C. Thompson, Vasilis Vasiliou

**Affiliations:** 1Department of Environmental Health Sciences, Yale School of Public Health, Yale University, New Haven, CT 06520, USA; caroline.johnson@yale.edu (C.H.J.); jayaprakash.golla@yale.edu (J.P.G.); evangelos.dioletis@yale.edu (E.D.); surendra.singh@yale.edu (S.S.); momoko.ishii@yale.edu (M.I.); georgia.charkoftaki@yale.edu (G.C.); david.c.thompson@yale.edu (D.C.T.); 2Department of Clinical Pharmacy, School of Pharmacy, University of Colorado Anschutz Medical Campus, Aurora, CO 80045, USA

**Keywords:** CRC, carcinogenesis, alcohol, acetaldehyde, ALDH1B1, CYP2E1, oxidative stress, DNA damage, immunosuppression, microbiome

## Abstract

**Simple Summary:**

Alcohol consumption is a leading cause of lifestyle-induced morbidity and mortality worldwide. It is well-established that there is an association between alcohol consumption and an increased risk of colorectal cancer. Long-term alcohol consumption causes a spectrum of liver diseases, including steatosis, hepatitis, and liver cancer, and is detrimental to many other organs. In the body, alcohol can be metabolized to chemicals that exhibit biological activity, such as acetaldehyde. The intracellular accumulation of these compounds can result in suppression of antioxidant defense systems, and alterations in DNA. In addition, they can elicit changes at the tissue level, leading to reductions in nutrient absorption, inflammation, and impairment of the immune system. Together, these effects may increase the risk of cancer in a variety of organs. This review discusses the mechanisms by which alcohol may promote colorectal cancer. It is anticipated that a clearer understanding of the mechanisms by which alcohol induces cancer will facilitate the development of more effective therapeutic interventions.

**Abstract:**

The etiology of colorectal cancer (CRC) is complex. Approximately, 10% of individuals with CRC have predisposing germline mutations that lead to familial cancer syndromes, whereas most CRC patients have sporadic cancer resulting from a combination of environmental and genetic risk factors. It has become increasingly clear that chronic alcohol consumption is associated with the development of sporadic CRC; however, the exact mechanisms by which alcohol contributes to colorectal carcinogenesis are largely unknown. Several proposed mechanisms from studies in CRC models suggest that alcohol metabolites and/or enzymes associated with alcohol metabolism alter cellular redox balance, cause DNA damage, and epigenetic dysregulation. In addition, alcohol metabolites can cause a dysbiotic colorectal microbiome and intestinal permeability, resulting in bacterial translocation, inflammation, and immunosuppression. All of these effects can increase the risk of developing CRC. This review aims to outline some of the most significant and recent findings on the mechanisms of alcohol in colorectal carcinogenesis. We examine the effect of alcohol on the generation of reactive oxygen species, the development of genotoxic stress, modulation of one-carbon metabolism, disruption of the microbiome, and immunosuppression.

## 1. Introduction

Colorectal cancer (CRC) is the third most common cancer diagnosed in the United States and the third leading cause of cancer-related deaths in men and in women. The etiology of CRC is complex, approximately 10% of individuals with CRC have predisposing germline mutations that result in familial syndromes, such as Lynch syndrome and familial adenomatous polyposis [1]. However, the development of sporadic CRC is likely due to a combination of environmental influences, genetic susceptibility, and immune response mechanisms. Environmental influences that have been linked to increased risk of CRC include diets that are rich in red, processed, grilled, and processed meats, fats, and diets low in fiber and folate. Other suspected risk factors include preexisting diseases (obesity, inflammatory bowel diseases, type 2 diabetes), smoking, disruption to circadian rhythms (night-shift work), and the presence and organization of pathogenic microbiota [2,3,4]. Chronic alcohol consumption has also been associated with a higher risk of developing CRC [5,6,7,8]; however, while it is clear that higher levels of consumption increase risk for many types of cancer, no safe threshold (i.e., the amount of alcohol that increases the cancer risk) has been established. In 2020, the global incidence of alcohol-induced colon, rectal, and liver cancers were 1.0, 0.7 and 1.7 per 100,000 people, respectively [9]. For context, the global incidence of other alcoholic liver diseases, such as hepatitis and cirrhosis were much higher, at 8.3 and 9.9 per 100,000 people, respectively [10]. Nevertheless, the increasing incidence of these alcohol-induced pathologies [11] emphasizes the need for understanding the mechanisms underlying their pathogenesis with the end goal of developing more effective therapeutic interventions.

The effects of ethanol in alcohol-containing beverages may be of particular relevance to the development of CRC. The metabolism of ethanol can generate genotoxic metabolites including acetaldehyde that cause DNA mutations and oxidative stress in the colorectum that can lead to cancer. In addition, the local ethanol-mediated effects on the colorectal mucosa and microenvironment can disrupt the microbiome and cause tissue inflammation. Ethanol may also increase the susceptibility of tissues to carcinogenesis by activating enzymes that enable the production of procarcinogens (such as N-nitrosamines), altering the metabolism and distribution of carcinogens, interfering with the repair of carcinogen-mediated DNA alkylation, suppressing the immune response to cancer stimulating cellular regeneration, and/or exacerbating dietary deficiencies [5,12]. This review aims to provide an analysis of cellular, genetic, metabolic, and microbial mechanisms by which ethanol is metabolized in the body and contributes to the development of CRC.

## 2. Alcohol Metabolites and CRC

The primary metabolites of ethanol are acetaldehyde and acetate. Ethanol is initially oxidized to acetaldehyde by alcohol dehydrogenase (ADH) enzymes in the liver (Figure 1). Catalase and cytochrome P450 2E1 (CYP2E1) enzymes also contribute to the oxidation of ethanol but to lesser extents. Acetaldehyde is further oxidized to acetate by aldehyde dehydrogenase (ALDH) isozymes. ALDH2 is the most active ALDH isozyme in acetaldehyde metabolism, followed by ALDH1B1 and ALDH1A1 [13,14]. Most of the acetate formed is subsequently converted to acetyl coenzyme A (CoA) outside of the liver. A byproduct of ethanol oxidation by ADH and ALDH enzymes is the reduction of NAD+ to NADH, which lowers the NAD+/NADH ratio. This, therefore, limits NAD+ availability. NAD+ is an essential cofactor required for continued ethanol oxidation, as well as to sustain essential metabolic pathways such as glycolysis, TCA cycle, and fatty acid oxidation. Therefore, NADH needs to be reoxidized in the mitochondria by the electron transport chain to regenerate NAD+ and increase the rate of alcohol metabolism [15].

The oxidative reactions that metabolize ethanol primarily occur in hepatocytes (liver cells). However, ethanol from alcohol consumption can reach the gastrointestinal (GI) tract, and oxidative enzymes are present in the intestinal mucosa that enable ethanol metabolism. In addition, ethanol can be metabolized by intestinal bacteria that have ADH enzymes, therefore enabling increased production of acetaldehyde in the GI tract [15]. It has also been shown that nonoxidative metabolism can also occur in the intestines. Ethanol reacts with membrane phospholipids to generate phosphatidylethanol, an abnormal phospholipid that increases intestinal cellular proliferation [17]. Ethanol can also react with free fatty acids to produce fatty acid ethyl esters, which cause cellular injury [18]. These nonoxidative pathways are not commonly used but could be relevant when the intestinal injury has occurred in chronic alcohol consumers [19].

Acetaldehyde is responsible for the majority of the adverse effects observed as a result of alcohol consumption and has been shown to increase the risk of developing various cancers [20]. Therefore, mechanisms of acetaldehyde production and reactivity are of utmost importance, particularly in the context of different organ sites. The mechanisms of acetate in colorectal carcinogenesis are still not clear, but it has been hypothesized to contribute to cancer growth by serving as a substrate for the synthesis of acetyl-CoA. This section further discusses the roles of acetaldehyde and acetate in the etiology and mechanisms of colorectal carcinogenesis.

### 2.1. Acetaldehyde and Genotoxicity

Acetaldehyde is a genotoxic compound that causes DNA strand breaks, sister chromatid exchanges, and gross chromosomal aberrations, all of which are procarcinogenic [21]. Acetaldehyde can promote the production of reactive oxygen species (ROS) and reactive nitrogen species (RNS). These reactive species include free radicals such as superoxide (O_2_^-^) and hydroxyethyl (CH3CHO^-^) and nonradicals such as hydrogen peroxide (H_2_O_2_). ROS can react with polyunsaturated fatty acids (PUFAs), which leads to the generation of lipid peroxidation products (malondialdehyde (MDA) and 4-hydroxy-2-nonenal (4-HNE) (Figure 2) [22,23,24]. MDA can produce specific point mutations and react with DNA to form additional adducts [25], whereas 4-HNE, can react with DNA bases to generate carcinogenic exocyclic etheno-DNA adducts; 1,N^6^-ethenodeoxyadenosine (εdA) and 3,N^4^-ethenodeoxycytine (εdC) [5,23,24]. Acetaldehyde can also directly interact with DNA to form a broad variety of adducts that have been linked to cancer [26,27]. The type of acetaldehyde adducts formed varies depending on the deoxyribonucleoside with which it interacts. For example, acetaldehyde is most reactive with deoxyguanosine (dG), and to a lesser extent, with deoxyadenosine (dA) and deoxycytidine (dC) [27,28]. The major acetaldehyde-derived DNA adduct produced in humans is a Schiff base, N^2^-ethylidene-2′-deoxyguanosine (N^2^-ethylidene-dG) adduct, which is highly stable when present in DNA. In *Aldh2*-null mice treated with 8% ethanol, high levels of N^2^-ethyl-dG (a molecule derived from N^2^-ethylidene-dG) were observed in the esophagus, tongue, and submandibular gland. This adduct has also been observed in individuals with alcoholism, suggesting N^2^-ethyl-dG may be a potential biomarker for acetaldehyde-induced DNA damage. Although numerous studies have demonstrated that acetaldehyde-derived DNA adducts are associated with carcinogenesis, systematic studies have not been conducted that evaluate the levels of DNA adducts in alcohol-related cancer. Therefore, the contribution of these adducts to alcohol-induced CRC remains to be established. Acetaldehyde can also activate proliferating cell nuclear antigen (PCNA), apurinic/apyrimidinic endodeoxyribonuclease 1 (APE1), heat shock protein family E 10 (Hsp10) member 1 (HSPE1), heat shock protein family B (small) member 1 (HSPB1), and heat shock protein family A (Hsp70) member 4 (HSPA4) [21,29,30]. Therefore, in addition to DNA adducts, acetaldehyde can activate proteins involved in DNA damage and repair, contributing to the development of cancer.

Acetaldehyde has also been shown to activate key oncogenic transcription factors including nuclear factor kappa-B (NF-κB) [32]. NF-κB regulates the transcription of pro-inflammatory cytokines, such as tumor necrosis factor α (TNFα), interleukin-6 (IL-6), and IL-1β. These cytokines have widespread effects on signaling pathways and are considered to be central players in CRC tumorigenesis [33,34]. An example of the link between these cytokines and CRC was shown in CRC cell lines, where IL-1β stimulated the upregulation of microRNA-181a causing decreased expression of tumor-suppressor phosphatase and tensin homolog (PTEN), it also caused the subsequent induction of cellular proliferation [33]. NF-κB activity is regulated by the inhibitor of the NF-κB kinase (IKK) complex [35]. It has been shown that mice lacking IKK-β expression in myeloid or intestinal epithelial cells were used to elucidate the role of NF-κB activation in tumorigenesis induced by AOM/dextran sulfate sodium (DSS) administration [36,37]. These mice exhibited a lower colon tumor load and smaller tumors than wild-type mice. Furthermore, the expression of cytokines and growth factors in the epithelium, lamina propria, muscularis mucosa, and submucosa indicated inflammatory processes that correlated with tumor progression (Table 1) [37]. In addition to proinflammatory cytokines, activation of the NF-κB pathway upregulates specific tumorigenic pathways such as the glycogen synthase kinase 3β/catenin beta-1/monocyte chemoattractant protein-1 (GSK3β/β-catenin/MCP-1) pathway that facilitates the progression of tumors and the development of distant metastases [38,39]. Alcohol studies in human CRC cell lines (HT-29, DLD-1, HCT116, SW480) show that ethanol can increase the mRNA and protein levels of MCP-1 and its target receptor C-C Motif chemokine receptor 2 (CCR2). Antagonism of CCR2 blocks ethanol-stimulated migration of CRC cells. Knock-down of MCP-1/CCR2 or β-catenin inhibits ethanol-stimulated CRC cell migration and loss of adhesion; cell migration and loss of adhesion correlate with tumor invasiveness and metastasis. This indicates that alcohol promotes cytosolic accumulation of β-catenin and its subsequent nuclear translocation, by inhibiting GSK3β activity and stimulation of MCP-1 gene promoter activity in a β-catenin-dependent manner [38].

Studies that have examined the link between acetaldehyde and CRC in vivo are limited. In rats, chronic oral ethanol administration and cyanamide (a potent ALDH inhibitor) increased colonic acetaldehyde and the incidence of rectal tumorigenesis [47]. A study in mice examined long-term ethanol administration (without the use of an additional tumor promotor) and observed an increase in plasma acetaldehyde levels and intestinal tumorigenesis [48]. Conversely, other studies have not shown an effect. In rats administered moderate levels of ethanol, gene expression of ALDH2 was significantly higher in those that were administered ethanol and had reduced inflammation and DNA damage [49]. Moreover, the efficient and rapid metabolism of acetaldehyde by mucosal cells of the colon acts as an important protective mechanism against the genotoxic effects of alcohol in the colon [50].

### 2.2. Acetate and Role in CRC

Acetate has been historically associated with a protective effect against the development of CRC [51]. Acetate is a short-chain fatty acid, which, in addition to butyrate and propionate, are produced by the microbial fermentation of dietary fiber and serve as energy sources for colonocytes. One of the main protective roles of acetate is to induce apoptosis of CRC cells by promoting lysosomal membrane permeabilization and release of cathepsin D [52]. However, recent studies have suggested that acetate may contribute to cancer growth by serving as a substrate for the synthesis of acetyl-CoA. Acetyl-CoA is a crucial metabolite, required for the synthesis of macromolecules such as fatty acids, that contribute to cell growth. Acetyl-CoA also serves as an epigenetic and post-translational modifier [53]. Under conditions of adequate oxygen and nutrient supply, proliferating tumor cells primarily use carbon and nitrogen derived from glucose and glutamine for cell growth. Under hypoxic or fasting conditions, the production of glucose-derived citrate and acetyl-CoA is impaired. The cells instead use alternate carbon sources such as acetate for the generation of acetyl-CoA and other required macromolecules to maintain cell growth [54]. In addition, glutamine can be used to generate cytosolic acetyl-CoA via reductive carboxylation of α-ketoglutarate to citrate and acetyl-CoA [55]. Hypoxia occurring in the tumors of aggressive cancers results in their reliance on acetate as a major source for the acetyl-CoA pool [53,54,56]. Therefore, it is possible that high levels of acetate produced from chronic alcohol consumption, can fuel the growth of CRCs by providing an alternate fuel for the production of acetyl-CoA, particularly under hypoxic or fasting conditions.

## 3. Alcohol-Metabolizing Enzymes and CRC

The regulatory actions of enzymes involved in alcohol metabolism have been linked to the development of CRC. ALDH1B1, which has a more minor role in alcohol metabolism, compared to ALDH2, has been shown to propagate CRC [57]. However, ethanol-induced CYP2E1 causes the production of mutagens that have been linked to CRC [58]. The roles of ALDH1B1 and CYP2E1 in CRC are described in this section.

### 3.1. ALDH1B1

The ALDH isoform ALDH1B1 is suggested to contribute to CRC growth, and studies have shown that suppression of ALDH1B1 expression in SW480 CRC cells inhibits spheroid formation in vitro and tumor formation in xenograft models in vivo [59]. In addition to having a lesser role in acetaldehyde metabolism, ALDH1B1 influences protein and mRNA expression of downstream targets of the Wnt/β-catenin, Notch, and PI3K/Akt-signaling pathways, which are involved in advanced stages of CRC tumorigenesis [16,59]. Moreover, ALDH1B1 is highly expressed in colon cancer cell lines and in tumors from patients with CRC; thus, it has been proposed as a promising biomarker for human CRC [14,57]. ALDH2, the main enzyme involved in acetaldehyde oxidation has only a weak association to the mechanisms of CRC as shown from epidemiologic studies [60,61]. However, when genetic polymorphisms occur to ALDH2 (and also ADH), acetaldehyde can accumulate and increase the risk of CRC [62,63].

### 3.2. CYP450s

Chronic ethanol consumption can promote the production of acetaldehyde through the induction of CYP2E1 in the liver and colon [64,65]. Chronic ethanol ingestion has been shown to increase CYP2E1 levels two- to threefold in mucosal cells of the small and large intestines of rats [66,67]. This metabolic reaction that generates acetaldehyde also produces large quantities of ROS and RNS, which exerts deleterious effects within cells and tissues [5,16,68,69]. It also leads to the enhanced activation of a variety of procarcinogens such as N-nitrosamines (N-nitrosodimethylamine and N-nitrosodiethylamine) [65,70,71], increased lipid peroxidation and free radical generation, modulation of cellular regeneration, and the development of nutritional deficiencies [72,73,74,75,76]. In addition, εdA and εdC levels correlate with CYP2E1 activity, increased cell proliferation, and consumption of alcohol [77]. 1,N^6^-ethenodeoxyadenosine and εdC have been identified in white blood cell DNA, urine, and liver samples from human subjects with alcohol-related diseases; adduct levels increased 10- to 100-fold in colon, liver, bile duct, esophagus, and pancreatic tissues from these individuals [78]. One study in patients showed that rectal biopsies from alcoholics (>60 g ethanol consumed/d) and controls (<20 g ethanol consumed/d) had various levels of CYP2E1 expression and etheno-DNA adducts in the rectal mucosa of both sets of patients [58]. Surprisingly, CYP2E1 expression was not significantly different between the control and alcoholic patients possibly due to additional induction of CYP2E1 by dietary or microbial products such as free fatty acids in the rectum. However, when all human subjects (both control and alcoholic) were combined, there was a correlation between CYP2E1 expression and etheno-DNA adduct production, showing that induction of CYP2E1, in general, is important in the production of these carcinogens in the rectum.

Genetic polymorphisms in CYP2E1 have been linked to increased susceptibility to ethanol-induced CRC [79]. The rs2031920 (*Rsa*I) polymorphism in the 5′-flanking promoter region of the *CYP2E1* gene, was shown to be associated with CRC risk [80]. This polymorphism is known to contain a potential binding site for hepatic transcription factor 1 (HNF-1), which regulates *CYP2E1* transcription [81,82]. Therefore, the rs2031920 polymorphism may lead to changes in CYP2E1 activity and contribute to the development of CRC by promoting the generation of carcinogens and oxidative stress.

## 4. Effects of Alcohol on One-Carbon Metabolism

As previously mentioned, ethanol can have a direct effect on carcinogenesis by producing a genotoxic metabolite (acetaldehyde), increasing ROS, and by producing acetate, which increases the generation of metabolites required for cancer growth. Another potential effect of ethanol in carcinogenesis is the modulation of one-carbon metabolism (1CM), which affects cancer cell metabolism and growth. Furthermore, 1CM is used to describe a set of complex biological reactions involved in the transfer of 1C moieties (e.g., a methyl group) from nutrients (including folate and its derivatives) to a diversity of macromolecules, such as DNA and proteins [83]. Notably, 1CM is of interest for the mechanisms of CRC, as it has roles in nucleotide synthesis, methylation, inflammation, oxidation, and energy metabolism, and is a target for CRC antifolate chemotherapy [84,85,86,87]. Below, we discuss the effect of alcohol on folate metabolism, 1CM enzymes, and metabolites that are involved in epigenetic regulation.

### 4.1. Folate Metabolism

Acetaldehyde has been shown to affect 1CM by lowering vitamin B_6_ (folate) bioavailability. Experiments in alcohol-fed rats have shown that an increase of colonic acetaldehyde coincided with a ~50% reduction in colonic mucosal folate, likely due to direct degradation of folate by acetaldehyde [83]. Several studies in humans and rats have further demonstrated that alcohol consumption results in lower folate and higher homocysteine serum levels, signifying poor absorption of folate in the gut [88,89]. A long-standing notion has been that dietary vitamin B supplementation may offset the reduced folate bioavailability observed in individuals with alcoholism and thereby return 1CM to normal physiological levels. However, meta-analyses of existing studies that focused on the potential association between dietary supplementation of 1CM substrates and the emergence of CRC have failed to show a beneficial prophylactic role of folate supplementation against CRC [90]. Prolonged folate intake by heavy alcohol consumers (≥15 g ethanol consumed/d) also does not appear to affect the incidence of CRC [91]. Therefore, further studies are needed to clarify whether folate/vitamin B_6_ fortification (and consequent restoration of any 1CM imbalances) is protective against CRC in the general population and in subjects that consume alcohol.

### 4.2. Genetic Polymorphisms in One-Carbon Metabolism and CRC Risk

Polymorphisms of genes encoding 1CM enzymes increase susceptibility to alcohol-induced CRC, by either lowering the activity of the enzyme or affecting its gene methylation status. These enzymes enable the production of metabolic intermediates and cofactors in 1CM that regulate DNA and RNA synthesis and DNA repair mechanisms, and most importantly, promote genome-wide DNA hypomethylation (one of the first steps in carcinogenesis). Methylenetetrahydrofolate reductase (MTHFR) genetic variants have been the focus of several epidemiologic studies involving alcohol consumption. MTHFR converts 5,10-methylenetetrahydrofolate to 5-methyltetrahydrofolate in 1CM and directs folate pools toward methionine production [92]. Decreased MTHFR activity has been linked to colorectal, breast, hepatocellular, head and neck, and esophageal cancers in alcohol drinkers [93,94,95,96,97,98]. MTHFR genotypes, specifically rs1801133 TT, rs1801131 AA, choline dehydrogenase (CHDH) rs12676 AA, and cystathionine-beta-synthase (CBS) rs234706 GG, are correlated with levels of alcohol intake and an increased risk of CRC [98]. Furthermore, a functional polymorphism of methionine synthase (*MTR*) appears to augment the colorectal adenoma potential of alcohol. The self-defense forces study on Japanese men showed that heavy (≥30 mL ethanol/d) (but not light (<30 mL/d)) alcohol consumers with the *MTR* 2756GG genotype had an increased risk of colorectal adenomas [99]. In liver tissue where the bulk of the methyl donor S-adenosyl-methionine (SAM) is generated, chronic alcohol exposure in rats was shown to reduce methionine synthase activity, leading to a decrease in the SAM to *S*-adenosylhomocysteine (SAH) ratio [100,101] (Figure 3). Such a disturbance in the SAM/SAH ratio can affect DNA methyltransferase (DNMT1) activity and DNA methylation patterns. A similar mechanism would be expected to occur in the colonic mucosa of alcohol consumers.

### 4.3. Alcohol and Methylation

Chronic alcohol consumption can shift the epigenomic profile of the individual to one that favors increased CRC incidence by disrupting intracellular storages of methyl groups that sustain the epigenome and 1CM [107]. Methylation of the 5′-cytosine-phosphate-guanine-3′ (CpG) island, commonly occurs at gene regulatory regions (including promoters) and is a fundamental epigenetic mechanism that controls gene activity [108]. Among the genes whose methylation status changes upon alcohol consumption, certain oncogenes are likely upregulated, and/or tumor suppressor genes are downregulated. For example, *DNMT*s are considered to be de novo methyltransferases due to their ability to methylate nucleotides [108]. Polymorphisms in the promoter region of *DNMT*, such as −149C > T, may enhance promoter activity and, in turn, lead to hypermethylation (and silencing) of genes (including tumor suppressor genes) [109]. Alcohol is a 1CM (and methylation) attenuator and may thus “neutralize” or alleviate the hypermethylating activity of the aforementioned *DNMT* genetic variants [110]. Hypomethylation and gene expression can be potentially affected by alcohol at the level of endogenous antioxidants (Figure 3). For example, the influence of alcohol may also extend to the transsulfuration pathway by depleting glutathione levels (and thereby increase ROS-induced stress) that ultimately further shifts 1CM away from SAM production [106].

There are multiple genes associated with CRC whose methylation, and expression is affected by alcohol-mediated changes to 1CM [106,111]. One of the newest candidate genes is mitogenic insulin-like growth factor 2 (*IGF2*), which has been previously implicated in CRC [112,113]. Hypomethylation of the differentially methylated region-0 (DMR0) of *IGF2* in alcohol consumers (≥15 g/d) is associated with an increased risk of CRC [114]. It is plausible that upregulation of the tightly controlled *IGF2* upon alcohol consumption plays a role in CRC.

Several cancer types (including CRC) that develop without heavy alcohol consumption rely on upregulation of the 1CM pathways to meet their increased biosynthetic needs, i.e., to replicate DNA and proliferate [115]. Therefore, while alcohol consumption appears to suppress 1CM and provide a favorable epigenetic landscape for CRC, it is still not fully understood how precancerous and developing cancer cells in alcohol consumers fulfill their tendency to upregulate 1CM and rapidly expand.

## 5. Effects of Alcohol on the Gastrointestinal System

Alcohol can also exhibit direct harmful effects on the GI system, causing microbial dysbiosis, increasing intestinal permeability and altering the balance of T cells and the immune response. These side effects when combined with the actions of alcohol metabolites, can produce a procarcinogenic environment giving evidence for alcohol-induced CRC.

### 5.1. The Microbiome

Chronic alcohol consumption can result in a dysbiotic (imbalance in microbial communities) colorectal microbiome. Studies have shown that alcohol use can decrease Bacteroidetes, Firmicutes, and butyrate-producing bacteria, and increase Proteobacteria and Actinobacteria [116]. These changes can lead to alterations in bacterial metabolism of dietary products and community structure, hyperpermeability of the colonic mucosa, translocation of bacterial products across the mucosa to the circulation, and chronic inflammation, all of which can predispose an individual to cancer [117].

In addition to the actions of ethanol on the diversity and abundance of microbiota in the colon, ethanol can itself undergo extrahepatic oxidation by the microbiome, the results of which results in high levels of acetaldehyde within the colorectum. The colorectal mucosal surface has an increased oxygen content compared to the lumen. It is believed that acetaldehyde production is elevated at the mucosal surface due to aerobic ethanol metabolism by ADHs and catalases from aerobes and facultative anaerobes, such as *Escherichia coli*. The production of acetaldehyde from these bacteria is in response to high levels of oxidative stress caused by chronic alcohol exposure. A study on acetaldehyde production in feces from individuals with alcoholism also observed that obligate anaerobes such as those from the *Ruminococcus* genus can generate acetaldehyde at high levels [118]. Due to the high levels of oxidative stress in the colon caused by alcohol exposure, it has been proposed that biofilms could form as a result [118]. The concentration of acetaldehyde-metabolizing bacteria in the biofilms could enhance acetaldehyde production, localizing and propagating its genotoxic effects. Colonic biofilms have been previously linked to an increased risk of CRC [2,3]. Their presence is associated with enhanced cell proliferation, increased activation of proinflammatory cytokines (IL-6, Stat3), and decreased E-cadherin; thus, chronic alcohol use may lead to the formation of procarcinogenic biofilms and could also cause a localized accumulation of acetaldehyde, further increasing the risk of cancer.

It has also been proposed that alcohol and its metabolites may interfere with the intestinal absorption or transport of folate [102,103]. Ethanol consumption may reduce folate availability by decreasing the population of folate-producing colonic bacteria, e.g., *Bifidobacterium* spp. Ethanol may instead increase the proliferation of anaerobes, which catabolize ethanol to acetaldehyde and therefore enhance the degradation of folate [5,65]. In addition, ethanol consumption decreases peristalsis/intestinal motility and thus could increase the exposure of folate to the degrading effects of acetaldehyde [119].

### 5.2. Intestinal Permeability

In addition to decreasing intestinal motility, chronic ethanol ingestion can result in further damage to the intestines by causing intestinal barrier dysfunction. This results in leaky gap junctions that facilitate the translocation of bacteria and their toxic metabolites (endotoxins) across the intestinal barrier. These processes result in intestinal inflammation by exposing immunogenic microbial materials to the lamina propria (where the immune cells reside) and therefore increase the susceptibility of epithelial cells to carcinogens in the colorectum [19]. The mechanism of ethanol-induced barrier dysfunction is not known but hypotheses generated from in vitro experiments include disruption of tight junction integrity through myosin light chain kinase activation with modulation of perijunctional action and myosin filaments [120], and NF-kB action causing F-actin cytoskeleton instability [19,121]. Another hypothesis is that ethanol upregulates a number of transcriptional regulators such as circadian clock proteins [122], and *Snail*, a gene involved in epithelial-mesenchymal transition, which is linked to metastasis [19,123]. Ultrasound of mucosa from heavy alcohol consumers also shows damage to epithelial cells, such as distorted mitochondria, which could contribute to barrier dysfunction [19,124].

## 6. Effects of Alcohol on the Immune System

Alcohol consumption is known to modulate both the innate and adaptive immune systems [125] and promote cancer progression [126]. However, direct information about how alcohol targets the immune system to enhance colon carcinogenesis is limited. Alcohol consumption may facilitate metastasis of CRC cells to neighboring organs, such as the liver; alcohol consumption may create an inflammatory liver microenvironment and inactivate natural killer and CD8^+^T cells that facilitate cancer development in the liver. Alcohol-administered mice had decreased numbers of natural killer cells and CD8^+^ T cells in the peripheral blood, compared to the control group treated with water [127], both of which are key to the antitumor surveillance function of the immune system. The lymphopenia induced by ethanol consumption in mice was also shown to be associated with a two- to fourfold decrease in mature B cells, CD4^+^ and CD8^+^ T cells [91]. Through such actions, alcohol could suppress the immune system and facilitate the spread of colon cancer cells to common secondary sites, such as the liver.

A potential mediator of the immune effects of alcohol consumption is chemokine (C-C motif) ligand 5 (CCL5/RANTES). This chemokine is expressed in colorectal tissue and acts as a chemoattractant for macrophages and T cells. It helps to sustain inflammation and promote malignant transformation. Importantly, CCL5/RANTES is upregulated in human CRC [128]. Studies in mice show that chronic ethanol ingestion increases CCL5/RANTES mRNA expression in distal colonic mucosa [129]. In regulatory T (T_reg_) cells isolated from tumor-draining lymph nodes of mice harboring CT26 (colon carcinoma cell) tumors, CCL5 treatment enhances T_reg_ cell-mediated CD8^+^ T cell apoptosis. In addition, tumor growth in *CCR5*-deficient mice is delayed and accompanied by reduced frequencies of both T_reg_ cells and apoptotic CD8^+^ T cells [130]. Therefore, it is plausible that alcohol can reduce the number of tumor-toxic CD8^+^ T cells via CCL5 upregulation, a process that would be anticipated to contribute to colorectal carcinogenesis.

CD4^+^ helper T cells can exert direct antitumor activity and provide the necessary signals for sustaining the aforementioned CD8^+^ T cell-mediated tumor suppression [131,132]. Activation and survival of CD4^+^ T cells are dependent on mitochondrial 1CM [131]. For example, suppression of the catabolic enzyme serine hydroxymethyltransferase 2 (SHMT2), part of the mitochondrial 1CM machinery, reduces proliferation and survival of CD4^+^ T cells in vivo and in vitro. By adversely affecting the 1CM pathways of immune cells, alcohol may impair the ability of the immune system to survey and detect alcohol-induced precancerous lesions in the colon. Moreover, a dampened immune response to intestinal bacterial that have translocated across the intestinal barrier may fuel CRC disease progression. Further research is required to determine the levels of alcohol consumption that activate or suppress the immune system and how these changes affect vulnerability to CRC development.

## 7. Conclusions

Chronic alcohol consumption is a convincing risk factor for CRC [5,6,7,8]. The mechanisms of alcohol-induced CRC are hypothesized to be through actions of ethanol metabolites—acetaldehyde, acetate, and alcohol-metabolizing enzymes. Acetaldehyde is a known carcinogen. It causes DNA damage via the production of ROS and RNS and can attenuate 1CM. It also causes dysbiosis and increased intestinal permeability, which leads to inflammation, immune disruption, and cancer. Genetic polymorphisms in ethanol-metabolizing enzymes such as ALDH2 can lead to a change in acetaldehyde production, whereas ALDH1B1 and ethanol-inducible CYP2E1 have effects on Wnt/β-catenin signaling and the production of procarcinogens, respectively. Acetate has more recently become associated with CRC via its metabolism to acetyl-CoA. Acetyl-CoA is an important metabolite in cancer; under hypoxic conditions, it is the primary source for building macromolecules that are required for cancer cell growth. Given that there are multiple mechanisms of alcohol-induced CRC, it is surprising that there is currently no safe threshold for the amount of alcohol that can be consumed safely. Further studies are needed to assess the link between alcohol consumption and CRC, and identification of those that may be at higher risk for CRC development.

## Figures and Tables

**Figure 1 cancers-13-04404-f001:**
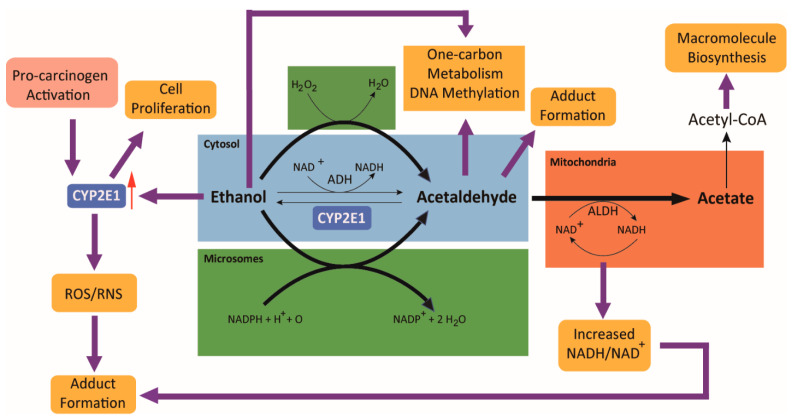
Role of ethanol and its metabolism in colorectal carcinogenesis. Ethanol is metabolized to acetaldehyde by alcohol dehydrogenase (ADH), cytochrome P4502E1 (CYP2E1), and catalase. Acetaldehyde is oxidized further to acetate primarily by acetaldehyde dehydrogenases (ALDHs). Acetaldehyde promotes the generation of reactive oxygen species (ROS)/reactive nitrogen species (RNS) and the formation of DNA and protein adducts, which contribute to the initiation and growth of colorectal cancer. It also modulates one-carbon metabolism and affects DNA methylation. Acetate contributes to the pool of acetyl CoA in hypoxic cancer cells and thereby sustains the synthesis of macromolecules required for cancer growth. Ethanol oxidation by CYP2E1 generates ROS/RNS and thus increases the production of DNA and protein adducts. Red arrow indicates the induction of CYP2E1 expression by ethanol and the activation of procarcinogens. Adapted from [16].

**Figure 2 cancers-13-04404-f002:**
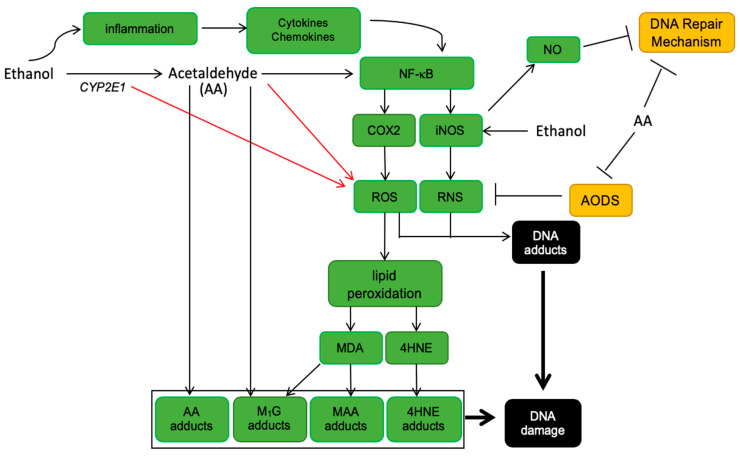
Effect of CYP2E1- and acetaldehyde-mediated generation of reactive oxygen species (ROS)/reactive nitrogen species (RNS) and DNA adducts on DNA damage. Induced expression of CYP2E1 results in the oxidation of ethanol, and the formation of acetaldehyde and reactive species (ROS/RNS). Acetaldehyde inhibits DNA repair mechanism and anti-oxidative defense system (AODS) [5]. Acetaldehyde and inflammation-derived cytokines activate nuclear factor kappa-B (NF-κB) in colon cells [31]. NF-κB stimulates the expression of cyclooxygenase 2 (COX-2) and inducible nitric oxide synthase (iNOS), which promote the additional formation of ROS, RNS, and lipid peroxidation products, such trans-4-hydroxy-2-nonenal (4HNE) and malondialdehyde (MDA), that interact with DNA bases to form adducts [16,24]. N-hydroxyethyl (HE) radicals and acetaldehyde (AA) adducts are formed during ethanol metabolism. Pyrimido-[1,2-a] purin-10(3H)-one (M_1_G) and MDA-acetaldehyde (MAA) adducts are generated by the reaction of DNA and protein with MDA [16,24]. Additional adducts are formed by the reaction of DNA with 4HNE. DNA adducts formed as a result of MDA and 4HNE are marked as in a rectangle. The generated DNA and protein adducts may react with DNA bases to form etheno-DNA adducts, which can promote DNA damage [5]. DNA repair mechanism refers to O6-guanine-methyltransferase and 8-oxo-guanine-DNA-glycosylateis, which are both inhibited by AA and nitric oxide (NO) [5]. Ethanol produces a nitric oxide system (iNOS) that generates NO [5]. Figure adapted from [16].

**Figure 3 cancers-13-04404-f003:**
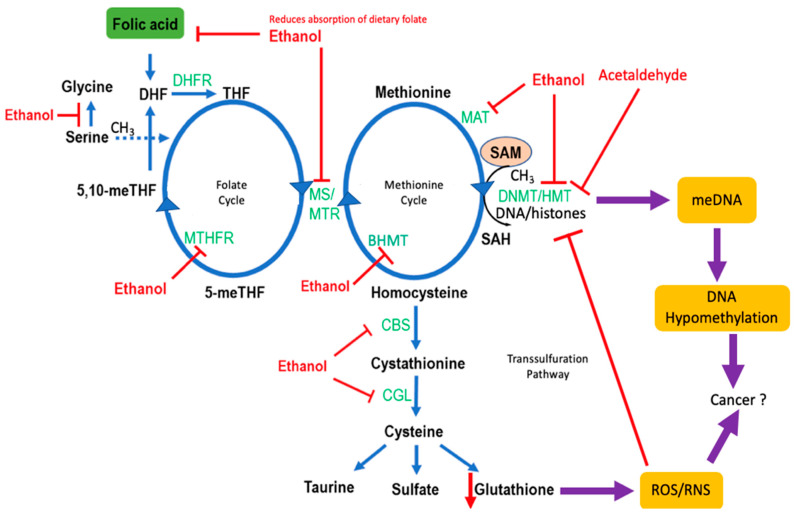
The influence of alcohol consumption on one-carbon metabolism. A schematic representation of the impact of ethanol on the main components of one-carbon metabolism (1CM) consists of three interlinked modules, folate and methionine cycles, and the transsulfuration pathway. Ethanol decreases the bioavailability of folate, a crucial “fuel” in 1CM. This is thought to occur by reducing the uptake of dietary folate and inhibiting enzymes involved in the folate cycle [102,103,104]. Ethanol inhibits methionine synthase (MS/MTR) [105] causing hyperhomocysteinemia and disrupts the production of S-adenosyl-methionine (SAM), a global methyl donor. Ethanol may also lower SAM production in a more direct manner by inhibiting methionine S-adenosyltransferase (MAT). DNA methyltransferases (DNMTs) and histone methyltransferases (HMTs) can also be inhibited by ethanol to result in DNA and histone hypomethylation, which would affect gene regulation and expression. This would have a resultant effect on decreasing cellular glutathione levels, augmented by the inhibitory effects of ethanol on cystathionine-β-synthase (CBS) and cystathionine-γ-lyase L (CGL) indicated with red arrow. The resulting elevated levels of ROS and RNS would increase the oxidative and nitrosative damage, further diverting 1CM away from SAM production to augment hypomethylation [106]. Genetic polymorphisms in methylenetetrahydrofolate reductase (MTHFR) can alter THF production [16]. Alcohol consumption can decrease THF production by inhibiting MTHFR [16]. Enzymes are shown in green text and metabolites in black text. DHF: dihydrofolate, THF: tetrahydrofolate, DHFR: dihydrofolate reductase, 5,10-meTHF: 5,10-methylenetetrahydrofolate, 5-meTHF: 5-methylenetetrahydrofolate, MTHFR: methylenetetrahydrofolate reductase, SAH: S-adenosyl-homocysteine. Figure adapted from [16].

**Table 1 cancers-13-04404-t001:** Molecular profile of CRC progression stages. During CRC progression, oncogenes are activated and tumor suppressor genes are inactivated with each successive stage. Putative tumor suppressors that are linked to metastasis are colored red. Growth factors pathways produce signaling molecules that promote tumor growth; key enzymes and signaling molecules noted in green for activated and blue for inhibited. *MSI* microsatellite instability, *MMR* mismatch repair, *CIN* chromosomal instability. *MLH1* MutL homolog 1, *PIK3CA* phosphatidylinositol-4,5-bisphosphate 3-kinase catalytic subunit alpha [40], *KRAS* K-ras, *BRAF* B-raf, *VIM* vimentin [41], *FN1* fibronectin 1 [42], *CDH2* cadherin 2 [43], *APC* adenomatous polyposis coli, *TP53* tumor protein p53, *BAX* bcl-2-like protein 4 or apoptosis regulator BAX, *SMAD4* small mothers against decapentaplegic homolog 4, *TGFBR2* transforming growth factor-beta receptor 2, *PTEN* phosphatase and tensin homolog [33], *E-cadherin* epithelial calcium-dependent adhesion [43], *CTNNA* alpha-catenin encoding gene [44], JUP gamma-catenin encoding gene [45], *TGF- β* transforming growth factor beta 1, *COX2* cyclooxygenase-2, *15-PGDH* 15-Hydroxyprostaglandin dehydrogenase, *EGFR* epidermal growth factor receptor, *PI3K* phosphoinositide 3-kinase. Adapted from [16,46].

CRC Progression Stage: Normal Cell → Adenomatous Polyps → High-Risk Adenoma → Cancer → Metastasis					
**Oncogenes**		MSI (MMR mutation) (MLH1 methylation) PIK3CA [40]	CIN (e.g., CDC4) KRAS, BRAF	PIK3CA	VIM [41] FN1 [42] CDH2 [43]
**Tumor suppressors**		APCβ -Catenin		TP53, BAX, SMAD4, TGFBR2PTEN [33]	E-cadherin [43] CTNNA (α-catenin) [44] JUP (γ-catenin) [45] TGF-β
**Growth factor pathways**	COX2	COX2 15-PGDH EGFR	EGFR	EGFR	EGFR
**Affected tissues**	None	Epithelium Lamina propria	Epithelium Lamina propria	Epithelium Lamina propria Muscularis mucosa Submucosa	Epithelium Lamina propria Muscularis mucosa Submucosa

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
