# Peer review of "Molecular Mechanisms of Alcohol-Induced Colorectal Carcinogenesis"

_cancers, 2021, doi:10.3390/cancers13174404_

Round 1

Reviewer 1 Report

The manuscript titled "Molecular mechanisms of alcohol-induced colorectal carcinogenesis" (cancers-1326194) submitted to Cancers, describes a very interesting and useful approach describing recent findings on the mechanisms of alcohol in colorectal carcinogenesis. Review is very well organized and written. Manuscript is divided properly and after short introduction, authors in very comprehensive way described Alcohol metabolites and CRC as well as Alcohol metabolites and CRC. Review is based on almost 130 references, concluded that there are multiple 
mechanisms of alcohol-induced CRC, and it is surprising that there is currently no safe threshold for the amount of alcohol that can be consumed safely. To sum up in my opinion manuscript is written in correct way, and it could be published in present form without any changes.

Author Response

We thank the Reviewer for taking time to review the manuscript.

Reviewer 2 Report

Title: Molecular mechanisms of alcohol-induced colorectal carcinogenesis

1.The incidence of alcohol induced CRC comparing with other alcohol induced comorbidity such as alcohol hepatitis, cirrhosis, and liver tumors?

2.Analyses of the interactions between 7 different factors about alcohol induced CRCs mentioned  in the manuscript.

English language and style are fine/minor spell check required

Author Response

We thank the reviewer for this suggestion and comment. We have now included the incidence of alcohol-induced CRC and alcoholic liver disease in the first paragraph of manuscript (in red).

In 2020, the global incidence of alcohol-induced colon, rectal, and liver cancers were 1.0, 0.7, and 1.7 per 100,000 people, respectively [9].  For context, the global incidence of other alcoholic liver diseases, such as hepatitis and cirrhosis were much higher at 8.3 and 9.9 per 100,000 people, respectively [10]. Nevertheless, the increasing incidence of these alcohol-induced pathologies [11] emphasizes the need for understanding the mechanisms underlying their pathogenesis with the end goal of developing more effective therapeutic interventions.